# Change in plasma cryptococcal antigen titers in Uganda among outpatients with advanced HIV disease from 2017 to 2022 with rollout of national cryptococcal screening

**Alexandra Poeschla**[1], **Abduljewad Wele**[2], **Elizabeth Nalintya**[3], **Biyue Dai**[2], **David B. Meya**[3,4], **David R. Boulware**[1], **Radha Rajasingham**[1]*

**1** University of Minnesota Medical School, Minneapolis, Minnesota, United States of America, **2** Department of Biostatistics, University of Minnesota School of Public Health, Minneapolis, Minnesota, United States of America, **3** Infectious Diseases Institute, College of Health Sciences, Makerere University, Kampala, Uganda, **4** Department of Medicine, School of Medicine, College of Health Sciences, Makerere University, Kampala, Uganda

* radha@umn.edu

## Abstract

### Introduction

Cryptococcal meningitis causes an estimated 19% of AIDS-related deaths globally and is the leading cause of meningitis in adults with HIV. Subclinical infection with cryptococcal antigen (CrAg) is detectable in plasma, and CrAg titer of>=1:160 is predictive of meningitis or death. We evaluated if plasma CrAg titer changed over time in Uganda with the expansion of national CrAg screening programs and antiretroviral (ART) access.

### Methods

We prospectively screened adults with advanced HIV disease (CD4 ≤ 200 cells/μl) for CrAg from 2017 through 2022 using the lateral flow assay and assessed median plasma CrAg titer.

### Results

From November 2017 to May 2022, 436 adults with advanced HIV disease had a positive plasma CrAg test. The median CD4 + cell count was 45 [IQR: 21,94] cells/μL, and median plasma CrAg titer was 1:80 [IQR: 10,1280]. Analysis of median quarterly CrAg titer from 2017–2022 demonstrated a non-statistically significant positive trend in CrAg titer (tau = 0.385, p = 0.086). There was a statistically significant decline in the percentage of participants taking ART at the time of screening (p < 0.001), with 58% reporting never having taken ART in 2022.

**Data availability statement:** Darlisha Williams Senior Program Manager University of Minnesota Coat.trial@gmail.com Data will be shared in a public repository in the Harvard Dataverse. DOI: https://doi.org/10.7910/DVN/IZJCOM.

**Funding:** R.R. is supported by the National Institutes of Health (grant number K23AI138851) and National Institute of Allergy and Infectious Diseases (grant number U01AI174978). D.R.B. is supported by the National Institutes of Health (grant number K24AI184270) and National Institute of Allergy and Infectious Diseases (grant number U01AI125003). The funders had no role in study design, data collection and analysis, decision to publish, or preparation of the manuscript.

**Competing interests:** The authors have declared that no competing interests exist.

## Conclusions

Despite expansion of CrAg screening and ART, median annual CrAg titers have not decreased between 2017 and 2022 in Uganda. Contrary to national reporting of expanded access to ART, our study population had higher rates of ART-naïve status over time, suggesting ongoing late presentation to care for people with advanced HIV disease. In addition, the presence of ART-experienced patients in our study population suggests challenges with treatment adherence and retention. Cryptococcosis persists, and despite public health efforts, people are not presenting to care earlier in their disease course. Continued refinement of CrAg screening programs is needed to reduce AIDS-related deaths.

## Background

Cryptococcosis causes an estimated 19% of AIDS-related deaths globally and is the leading cause of meningitis in adults living with HIV in Africa [1]. Subclinical infection with cryptococcal antigen (CrAg) in the bloodstream is a risk factor for meningitis and death [2]. The World Health Organization (WHO) recommends CrAg screening for adults with a CD4 cell count <100 cells/μL and pre-emptive fluconazole treatment for people identified as plasma CrAg-positive [3]. At least 19 African countries have now implemented national CrAg screening programs for people with advanced HIV disease [4]. In Uganda, the national CrAg screening program for adults with a CD4 cell count <100 cells/μL was implemented in 2016 [5], with a conditional recommendation to expand this screening to all people with CD4 cell counts <200 cells/μL. CrAg screening programs identify people earlier in their course of cryptococcal disease and result in improved survival [3].

In asymptomatic adults with HIV, a blood CrAg titer of ≥1:160 is associated with increased risk of meningitis and death [2]. Conversely, people with CrAg titers <1:160 who are treated with fluconazole have survival that is equivalent to that of CrAg-negative persons [6,7].

Antiretroviral therapy (ART) coverage in Uganda has ranged from 67% in 2017 to 80% in 2022 [8]. The objective of this study is to evaluate whether plasma CrAg titer has changed over time in the context of the rollout of the national CrAg screening program and expanded access to ART in Uganda [5,9]. We hypothesize that CrAg titer has decreased annually, as people are presenting to care earlier in their disease course before onset of fulminant meningitis.

## Methods

### Study setting and patients

We prospectively screened adults with advanced HIV disease (CD4 ≤ 200 cells/μl) for CrAg in the plasma using the lateral flow assay (IMMY, Norman, Oklahoma, USA) to identify potential participants for clinical trials (ClinicalTrials.gov: NCT03945448, NCT03002012, NCT01535469) [6,7,10]. Those who received CD4 testing had either

newly enrolled into care, were returning to care, or had ceased taking ART. If CD4 count was identified as less than 200 cells/µL, CrAg testing was performed. If CrAg was detected, titers were performed prospectively, in real time (not on frozen samples), using serial dilutions of the CrAg lateral flow assay at a central laboratory at the Infectious Diseases Institute in Kampala, Uganda. Participants in this analysis were adults with advanced HIV disease who received a CrAg-positive result in plasma with an associated titer performed. Screening occurred at outpatient HIV clinics in the Kampala and Wakiso districts and the Masaka Medical Research Council field station as part of standard medical practice. On May 26 2022, our clinical trial stopped enrolling participants with low CrAg titers (≤1:80). Thus, we did not include participants screened after May 31, 2022 due to possible bias in screening practices favoring high titer participants.

## Statistical methods

Baseline characteristics including age, sex, CD4 count, ART status, time on ART, and symptoms at presentation were collected and analyzed for study participants. We assessed the median and interquartile range (IQR) plasma CrAg titer quarterly and annually between November 15, 2017 through May 31, 2022. We used Fisher exact tests to compare the proportion with a high CrAg titer quarterly and annually. Mann-Kendall trend test and Sen's estimate were used to evaluate the trend of quarterly median values of blood CrAg titer over time (2017–2022) using Rstudio: Integrated Development Environment for R, 2024.Version 12.0.

## Institutional review board and informed consent

A waiver of consent was obtained as CrAg screening is part of routine HIV care. Persons eligible for clinical trials provided written informed consent. The studies were conducted in accordance with the Declaration of Helsinki, and international ethical guidelines for biomedical research involving human subjects. They were conducted in compliance with the relevant government regulations for Uganda, and the U.S., international, and institutional research policies and procedures. All study protocols (ClinicalTrials.gov: NCT03945448, NCT03002012, NCT01535469) were approved by the Institutional Review Board of the Uganda National Council for Science and Technology, Uganda National Drug Authority, and the University of Minnesota.

## Results

Between November 2017 and May 2022, 587 Ugandan adults with advanced HIV disease were identified with cryptococcal antigenemia. Of the 587 CrAg-positive participants, 436 had CrAg titers performed and were included in this analysis. The median age of participants was 36 [IQR: 30,42] years, and 49% of participants were female (Table 1). The median CD4+cell count was 45 [IQR: 21,94] cells/µL, and median plasma CrAg titer was 1:80 [IQR: 1:10,1:1280]. There was a statistically significant decrease in the percentage of participants taking ART at study enrollment over time (p<0.001).

Median and quartiles of CrAg titer are summarized and results from trend analysis are reported in Fig 1. A tendency toward an increased trend in the median CrAg titer is indicated by the positive Kendall's tau coefficient, although the trend is not statistically significant (tau=0.385, p=0.086). Likewise, a median $\log_2$ fold change of 0.18 each quarter is shown by the median trend's slope, suggesting a slight increase in CrAg titer over time.

## Discussion

In this analysis of Ugandan adults with advanced HIV disease and cryptococcal antigenemia, median annual CrAg titer did not decrease between 2017 and 2022. In contrast, there was a modest positive trend in median annual titer, though not statistically significant. We had hypothesized that the CrAg titer would decrease given the availability of ART and roll-out of the national CrAg screening program in Uganda. Since the implementation of PEPFAR in Uganda in 2004, access to ART has steadily increased [9]. ART coverage in Uganda has ranged from 67% in 2017 to 80% in 2022 [8]. Despite this improvement, no change has been noted in CrAg titer over this time. Notably, in our cohort, 58% were ART naïve in

**Table 1. Baseline characteristics of participants by year of cryptococcal antigen screening.**

| | N with data | Overall | 2017−18 (N=24) | 2019 (N=96) | 2020 (N=103) | 2021 (N=153) | 2022 (N=60) | P-value |
|---|---|---|---|---|---|---|---|---|
| Age, years | 436 | 36 [30,42] | 33 [30,37] | 35 [28,42] | 36 [30,42] | 36 [30,43] | 36 [31,42] | 0.348 |
| Female sex | 436 | 214 (49%) | 13 (54%) | 54 (56%) | 45 (44%) | 73 (48%) | 29 (48%) | 0.471 |
| CD4+cell count, cells/μL | 316 | 45 [21,94] | 39 [12,85] | 32 [13,93] | 44 [18,98] | 59 [26,109] | 45 [23,71] | 0.203 |
| Symptoms of meningitis | 436 | 182 (42%) | 7 (29%) | 43 (45%) | 48 (47%) | 66 (43%) | 18 (30%) | 0.168 |
| Plasma CrAg titer | 436 | 80 [10,1280] | 10 [5,40] | 80 [15,1920] | 80 [10,2560] | 160 [10,1280] | 80 [20,960] | 0.004 |
| Plasma CrAg titer <1:160 | 436 | 234 (54%) | 20 (83%) | 53 (55%) | 57 (55%) | 73 (48%) | 31 (52%) | 0.027 |
| Antiretroviral therapy (ART) status | 436 | | | | | | | |
| Currently on ART | | 213 (49%) | 16 (67%) | 67 (70%) | 56 (54%) | 53 (35%) | 21 (35%) | <0.001 |
| Never on ART | | 201 (46%) | 8 (33%) | 25 (26%) | 44 (43%) | 89 (58%) | 35 (58%) | |
| Previously on ART | | 22 (5%) | 0 (0%) | 4 (4%) | 3 (3%) | 11 (7%) | 4 (7%) | |
| Time on ART, months | 210 | 2 [0,8] | 8 [1,44] | 2 [0,11] | 1 [0,5] | 1 [0,7] | 3 [1,10] | 0.240 |

Data are presented as n (%) or median [IQR]. Fisher exact test was used to compare proportions and Kruskal-Wallis test was used for medians. Symptoms of meningitis included one or more of the following: headache, nuchal rigidity, photophobia, confusion, focal neurologic deficits, mania, or fever.

Abbreviations: ART, antiretroviral therapy; CrAg, cryptococcal antigen; IQR, interquartile range.

both 2021 and 2022, the highest rates over our study period, suggesting ongoing late presentation to care. There was a statistically significant decline in prevalence of ART coverage at enrollment over time. Thus, there is a disconnect between the national reports of improvement in ART coverage, and ART coverage of this population with advanced HIV disease. Additionally, the presence of participants in our cohort that were ART-experienced and yet still presented with a CD4 count less than 200 and a CrAg-positive plasma test suggests possible challenges with treatment adherence, interruption of treatment, virologic failure, or incomplete immune recovery.

Despite the lack of decrease in outpatient CrAg titers, we previously reported a lower severity of cryptococcal meningitis among hospitalized patients over this same time period [3,11]. Those with meningitis that were referred to the hospital after a positive CrAg screening test have 49% lower in-hospital mortality, underscoring the importance of these screening programs [3].

Few other studies have evaluated change in CrAg titers over time in HIV endemic settings. In 2016, South Africa became the first country to implement a reflex CrAg screening program for all patient samples found to have a CD4 count less than 100 cells/μL. An analysis of CrAg trends in South Africa from 2017–2019 did not find any significant change in CrAg positivity [12]. However, this study did not specifically analyze CrAg titer trends among positive results.

Although plasma CrAg titer screening is rapid, cost-effective, and can be done at the bedside [13], significant barriers to CrAg screening persist, and are exacerbated in low-resource settings. First, health systems must have timely CD4 testing in place, or if CD4 testing is not available, CrAg screening should be considered if the prevalence of advanced HIV disease exceeds 30% [3]. Of note, the adoption of the WHO "Treat All" strategy in Uganda, which recommends ART initiation for all people with HIV regardless of CD4 count, has led to diminished rates of CD4 testing. Baseline CD4 assessment for adults initiating ART decreased from 73% to 21% between 2013 and 2018 [14]. The "Treat All" strategy, while meant to reduce barriers to ART initiation and reduce morbidity and mortality among people with HIV, raises concerns that people with advanced HIV may not receive opportunistic infection screening and prophylaxis and could be a factor in the rise in CrAg titer over time even with increased access to ART.

If CD4 testing is done, and CD4 count returns <100 cells/μL, CrAg screening should be performed reflexively without relying on a clinician's order. Such reflexive testing results in a higher proportion receiving CrAg screening but is not implemented everywhere [15]. Of note, a study of 15 public health facilities in Uganda showed that mean adherence to

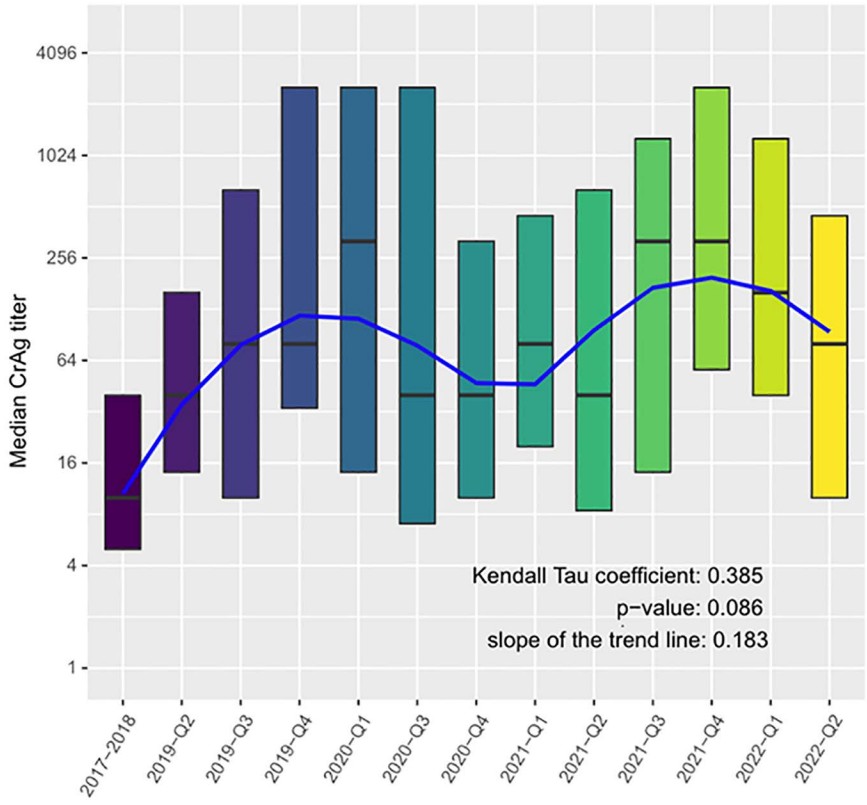

**Fig 1. Trend in baseline median CrAg titer from 2017-2022 by quartiles.** The blue line represents the estimated Sen's trend.

guidelines for screening and management of cryptococcal meningitis was low, with only 19% of those eligible for CrAg screening actually screened [16]. Furthermore, while the CrAg lateral flow assay is a simple, reliable tool [13], errors occur where the positive control is used inadvertently resulting in false positive results. Additionally, a small number of reports have documented the prozone or "hook" effect, in which exceptionally high levels of cryptococcal antigen prevents the binding of antigen-antibody complexes to the strip-anchored antibody, leading to false-negative results; in such cases with a high suspicion for cryptococcal meningitis but a negative result, the sample should be diluted and re-tested [17]. A positive plasma CrAg result should trigger expedited assessment of symptoms and consideration for lumbar puncture [15]. If meningitis is not present, fluconazole preemptive therapy is needed to reduce risk of progression to meningitis and/ or death. Thus, there are many seemingly simple steps for CrAg screening, though successful implementation requires investment in public health systems and training of healthcare staff.

There are limitations to this analysis. We present here data on individuals screened for clinical trials; we did not collect demographic or CD4 data on screened individuals. Our inclusion period overlaps with the Covid-19 pandemic; Uganda enforced strict lockdowns, and a higher burden of advanced HIV disease was seen after easing of these restrictions [18]. It is unclear how Covid-19 restrictions may have affected the trend in CrAg titer, but it is possible that Covid-19 lockdown measures caused patients to delay presenting to care, leading to patients presenting with more advanced disease. If volume of screening changed over time, trends may reflect better documentation of titer or CrAg result, rather than worsening epidemiology. Additionally, while 587 participants were CrAg-positive, only 436 had titers analyzed; it is unknown if persons without CrAg titers performed were a similar group to persons included in this analysis. It is possible that people excluded were more sick, and did not have a titer performed because they were triaged to the hospital for meningitis

evaluation. Finally, our findings are restricted to the outpatient population. This study took place in the Kampala, Wakiso, and Masaka districts of Uganda, and these findings may not reflect sites more distant from the city center. Further analysis of CrAg titer trends in other countries with a high burden of HIV would provide more insight into whether national screening programs result in earlier identification of CrAg-positive individuals, and lower plasma CrAg titer.

Despite nationally reported expansion of ART access and CrAg screening programs in Uganda, there has been no reduction in CrAg titer, a marker for meningitis and death. Our study population of people with advanced HIV disease had higher rates of ART-naïve status over time, suggesting ongoing late presentation to care. In addition, the presence of ART-experienced patients in our cohort suggests ongoing challenges with treatment adherence and retention. Continued efforts are needed to improve public health infrastructure and train healthcare workers on the implementation of CrAg screening to reduce and ultimately eliminate AIDS-related deaths.

## Acknowledgments

The authors would like to gratefully acknowledge the contributions of the study participants, providers, and research staff.

## Author contributions

**Conceptualization:** Alexandra Poeschla, Radha Rajasingham.

**Data curation:** Abduljewad Wele, Biyue Dai.

**Formal analysis:** Abduljewad Wele, Biyue Dai.

**Funding acquisition:** David B Meya, David R Boulware, Radha Rajasingham.

**Investigation:** Elizabeth Nalintya, David B Meya, David R Boulware, Radha Rajasingham.

**Methodology:** Alexandra Poeschla, Abduljewad Wele, Biyue Dai.

**Project administration:** Elizabeth Nalintya, David R Boulware, Radha Rajasingham.

**Resources:** Elizabeth Nalintya, David B Meya, Radha Rajasingham.

**Software:** Abduljewad Wele, Biyue Dai.

**Supervision:** Elizabeth Nalintya, David B Meya, David R Boulware, Radha Rajasingham.

**Validation:** Radha Rajasingham.

**Visualization:** Alexandra Poeschla, Abduljewad Wele, Biyue Dai.

**Writing – original draft:** Alexandra Poeschla.

**Writing – review & editing:** Alexandra Poeschla, Radha Rajasingham.

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
