## [Decision Letter · Decision Letter 0]

5 Mar 2026

PONE-D-25-65883Change in plasma cryptococcal antigen titers in Uganda among outpatients with advanced HIV disease from 2017 to 2022 with rollout of national cryptococcal screeningPLOS One

Dear Dr. Rajasingham,

Thank you for submitting your manuscript to PLOS ONE. After careful consideration, we feel that it has merit but does not fully meet PLOS ONE’s publication criteria as it currently stands. Therefore, we invite you to submit a revised version of the manuscript that addresses the points raised during the review process.

We look forward to receiving your revised manuscript.

Kind regards,

Felix Bongomin, MB ChB, MSc, MMed, FECMM

Academic Editor

PLOS One

**Journal Requirements:**

“R.R. is supported by the National Institutes of Health (grant number K23AI138851) and

National Institute of Allergy and Infectious Diseases (grant number U01AI174978). D.R.B. is supported by the National Institutes of Health (grant number K24AI184270) and National Institute of Allergy and Infectious Diseases (grant number U01AI125003). “

5. In the online submission form you indicate that your data is not available for proprietary reasons and have provided a contact point for accessing this data. Please note that your current contact point is a co-author on this manuscript. According to our Data Policy, the contact point must not be an author on the manuscript and must be an institutional contact, ideally not an individual. Please revise your data statement to a non-author institutional point of contact, such as a data access or ethics committee, and send this to us via return email. Please also include contact information for the third party organization, and please include the full citation of where the data can be found.

6. PLOS requires an ORCID iD for the corresponding author in Editorial Manager on papers submitted after December 6th, 2016. Please ensure that you have an ORCID iD and that it is validated in Editorial Manager. To do this, go to ‘Update my Information’ (in the upper left-hand corner of the main menu), and click on the Fetch/Validate link next to the ORCID field. This will take you to the ORCID site and allow you to create a new iD or authenticate a pre-existing iD in Editorial Manager.

Reviewers' comments:

Reviewer's Responses to Questions

**Comments to the Author**

1. Is the manuscript technically sound, and do the data support the conclusions?

Reviewer #1: Yes

Reviewer #2: Yes

Reviewer #3: Yes

Reviewer #4: Partly

Reviewer #5: Partly

2. Has the statistical analysis been performed appropriately and rigorously? 

Reviewer #1: Yes

Reviewer #2: Yes

Reviewer #3: Yes

Reviewer #4: No

Reviewer #5: Yes

3. Have the authors made all data underlying the findings in their manuscript fully available?

Reviewer #1: Yes

Reviewer #2: Yes

Reviewer #3: Yes

Reviewer #4: No

Reviewer #5: Yes

4. Is the manuscript presented in an intelligible fashion and written in standard English?

Reviewer #1: Yes

Reviewer #2: Yes

Reviewer #3: Yes

Reviewer #4: Yes

Reviewer #5: Yes

5. Review Comments to the Author

Reviewer #1: Authors indicated that data fully available upon request. The manuscript is written clearly and is technically sound. Interesting to note that there were about equal proportions of current on ART and never on ART among these PLHIV with AHD.

Reviewer #2: Refer to the attached comments. This is a well written article that generates important information on CrAg titers in a Sub-Saharan setting. It may be outside the scope of this study, but it would be valuable to add how many CrAg positive participants received antifungal treatment and a lumbar puncture.

Reviewer #3: The authors report an important analysis of cryptococcal antigen (CrAg) titers in Uganda during a period of expanded CrAg screening and ART scale-up. The observation that titers have not declined, and may even be increasing, despite these efforts is both notable and concerning. The authors convey a clear message that HIV remains a major public health challenge, the burden of advanced HIV disease (AHD) persists, and sustained efforts toward early detection and prevention remain essential.

I wonder whether increased screening efforts may have enabled the identification of a higher-risk population, particularly ART-experienced yet non-adherent patients (as possibly suggested in Table 1), which could partially explain the lack of overall improvement in findings. Despite the higher titers observed, it may be encouraging that these patients are being identified in the outpatient setting, reinforcing the importance of continued screening efforts.

The manuscript is well written, and I have no major concerns regarding the study design or analysis. The comments below are mostly to clarify several points and strengthen interpretation of the findings.

1. The authors state (Line 73): “In this analysis were asymptomatic adults with HIV who received a CrAg-positive result in plasma with an associated titer performed.” However, in Table 1, “symptoms of meningitis” are reported in a substantial proportion of the cohort (e.g., 42% overall).

It would be helpful to clarify this apparent discrepancy. Does “asymptomatic” refer specifically to the absence of meningitis symptoms at the time of screening, with symptoms assessed later during clinical evaluation?

2. The authors state (Lines 49 and 68): “The World Health Organization (WHO) recommends CrAg screening for adults with a CD4 cell count <100 cells/μL…” and “We prospectively screened adults with advanced HIV disease (CD4 <200 cells/μL)… to identify potential participants for clinical trials…”

It may be helpful to clarify which CD4 cutoff (100 vs 200) was applied for CrAg screening in the cohort included in this analysis. Additionally, please specify whether any further inclusion or exclusion criteria were used, given that participants were approached for clinical trials, and confirm whether these criteria remained consistent between 2017 and 2022.

Minor comments:

3. If available, it would be informative to report whether mortality changed over time. If mortality declined despite stable or increasing titers, this could suggest improvements in clinical management or the effectiveness of earlier identification and treatment following screening.

4. Were there changes in routine CD4 testing practices (e.g., greater reliance on viral load monitoring) that may have delayed recognition of advanced immunosuppression and, consequently, eligibility for CrAg screening? This is more speculative but may help contextualize the findings.

Reviewer #4: Review

Change in plasma cryptococcal antigen titers in Uganda among outpatients with advanced HIV disease from 2017 to 2022 with rollout of national cryptococcal screening

A very interesting concept for a manuscript, however the median quarterly CrAg titer from 2017-2022 demonstrated a non-statistically significant positive trend in CrAg titer (tau=0.385, p=0.086). Would any other factors affect enrolment in care and CrAg titres such as education, income/employment, marital status, sexual orientation etc? It seems like the COVID-19 pandemic and lockdown measures may have played a role in reducing patient enrolment/follow-up in care and access to antiretroviral therapy, probably resulting in patients presenting with more advanced disease? This should be discussed

Methods

Study setting and patients

Page 4 Line 68-69 : We prospectively screened adults with advanced HIV disease (CD4<200) for CrAg in the plasma using the CrAg lateral flow assay…..

Were these patients newly enrolled in care? Defaulted and returned to care?

Page 4 Line 72-73: Participants in this analysis were asymptomatic adults with HIV who received a CrAg-positive result…..

Here the adults were said to be asymptomatic but in Table 1, a number of participants had symptoms of meningitis. How is this explained?

Page 4 Line 76-78 : In May 2022, our clinical trial began only enrolling participants with high titers (>=1:160). Thus, we did not include participants screened after May 31, 2022 due to possible bias in screening practices ….

It would have been better not include patients screened in May 2022 as the clinical trial began in May 2022. A better cut-off may have been April 30, 2022

Results

In Table 1, p values of 0.004 and 0.027 were obtained for the Plasma CrAg titer and Plasma CrAg Titer<1:160 respectively, but these have not been adequately explained

Was a multivariable analysis conducted?

Discussion

Page 7 Line 140: First, health systems must have timely CD4 testing in place …

What method of CD4 testing is available in Uganda ? Is Visitect available

Page 8 Line 146-147: .. errors occur where the positive control is used inadvertently resulting in false positive results. …

It is difficult to understand this concept

Page 7 Line147-150: Additionally, a small number of reports have documented the prozone or “hook” effect, in which exceptionally high levels of cryptococcal antigen prevents the binding of antigen-antibody complexes to the strip-anchored antibody, leading to false-negative results.

How does one suspect the prozone effect among patient samples? Among cases with a high suspicion of cryptococcal meningitis and a negative CSF CrAg LFA, should serial dilutions be performed routinely?

Page 8 Line 151 : …consideration for lumbar puncture

Is it a simple procedure to refer patients to hospital with suspected cryptococcal meningitis an have a lumbar puncture done in Uganda? Staffing, equipment etc to conduct a lumbar puncture are readily available?

Page 8 Line 162-163 : Due to the COVID-19 pandemic “enforced strict lockdowns, and a higher burden of advanced HIV disease was seen after easing of these restrictions”

With a higher burden of advanced disease, one may expect a higher CrAg titre after easing of COVID-19 restrictions and this may affect the hypothesis that CrAg titer has decreased annually, as people are presenting to care earlier in their disease course before onset of fulminant meningitis

Page 8 Line 163-164: It is unclear how Covid-19 restrictions may have affected antiretroviral therapy at enrollment or trend in CrAg titer.

Could it be hypothesized that the COVID-19 lockdown measures may have affected patient enrolment/follow-up in care and access to antiretroviral therapy, probably resulting in patients presenting with more advanced disease?

Reviewer #5: This study aims to describe the temporal trend in blood CrAg titre for asymptomatic adults screened (or enrolled?) in a clinical trial in Uganda over a 5 year period. The authors hypothesize that CrAg titre at enrolment will increase during the five year period due to expanded access to ART and the existence of a CrAg screening programme.

Although the report is well written and clear, there are some missing details and the rationale for the study does not hold well. It is not clear whether the study has power to detect a temporal trend in CrAg titre over such a short period, or why the authors hypothesise that there should be a change during this time in asymptomatic individuals. I suggest the focus on temporal trend be re-assessed and the report rewritten to focus on a descriptive analysis of this cohort.

Background

- The background focusses on asymptomatic CrAgaemia but the authors hypothesize that blood CrAg titre at presentation to care will have decreased over a 5 year period due to people presenting earlier in disease course before onset of fulminant meningitis. This does not justify the study, since they enrolled only asymptomatic individuals.

- It is not clear why the authors would expect asymptomatic pts to present earlier because of the CrAg screening programme. This would potentially be impacted by other changes in HIV care leading to patients with low CD4 counts being detected earlier and engaging with care. The existence of CrAg screening in itself would have no impact on baseline CrAg titre.

- If improvements in HIV care and ART access during the study period drive the hypothesis, this should be expanded and explained with evidence presented.

Study setting and patients

- Please expand the methodology – how were symptomatic participants excluded? Were LPs performed? Sample freeze-thaw? Titre method? Were there any exclusions e.g. specific symptoms/signs, seizures? Previous CM? duration of antifungal treatment? Previous cryptococcal antigenaemia?

- Not clear why stopped enrolling participants after 2022 since the study relied on routine blood CrAg screening Were the study participants only those enrolled in the trial – be explicit.

- Demographic and CD4 data of patients excluded from enrolment should be presented and compared with those enrolled to show any bias. Please present these as an appendix.

Statistical methods

Justify sample size in relation to study aims – does this study have power to detect expected temporal trends in blood CrAg titre over a 5 year period?

Discussion

The discussion focusses on data from other studies and CrAg screening in general, which does not relate specifically to the study. The statement that ‘despite lack of decrease in outpatient CrAg titers, we previously reported a lower severity of CM among hospitalised patients’ should be explained. The trend in lower severity of CM is not likely to be related to CrAg titre at the time of presentation in asymptomatic individuals. The fact that baseline CrAg titres are static does does indicate that screening vs no screening is leading to initiation of treatment for CM earlier by detecting patients with early symptomatic disease (or prompting more LPs in asymptomatic individuals).

I would suggest a re-focus of discussion around ART trends detected in this study. The authors state that ‘it is unclear if the decrease in ART affected trend in CrAg titre’ – is there a relationship between ART and titre in this cohort? This would draw out more interesting discussion from this study.

6. PLOS authors have the option to publish the peer review history of their article (what does this mean?). If published, this will include your full peer review and any attached files.

Reviewer #1: No

Reviewer #2: **Yes:** Naseem Cassim

Reviewer #3: No

Reviewer #4: No

Reviewer #5: No

---

## [Author Response · Author response to Decision Letter 1]

7 Apr 2026

Reviewer #1: Authors indicated that data fully available upon request. The manuscript is written clearly and is technically sound. Interesting to note that there were about equal proportions of current on ART and never on ART among these PLHIV with AHD.

Response: Thank you for your feedback. We find this interesting as well.

Reviewer #2: This is a well written article that generates important information on CrAg titers in a Sub-Saharan setting. It may be outside the scope of this study, but it would be valuable to add how many CrAg positive participants received antifungal treatment and a lumbar puncture.

Response: Thank you. Unfortunately, we do not have this information about the participants. We only have data on antifungal treatment and incidence of lumbar puncture for participants who were enrolled, which is a subset of the overall cohort. In this manuscript we summarize CrAg titers among people screened, and we do not have further diagnostic and treatment data available for all individuals.

Comments to authors:

Major issues

In the Introduction, please provide some perspective of the ART coverage for Uganda for the study period. This would be important as the data seems to indicate increasing ART coverage among PLHIV. This could then be contrasted to the lower ART coverage for the CrAg positive participants.

Response: Thank you. We have added the following to lines 67-68: “Antiretroviral therapy (ART) coverage in Uganda has ranged from 67% in 2017 to 80% in 2022.

Table 1 is confusing as it is not clear whether the median CD4 count or the number of participants are reported by year. It would be better to split this into two tables, one with frequencies and a second with the median and IQR values.

Response: In table 1, data are presented as n (%) or median [IQR]. As median CD4 count is reported in the table as x[y,z], this shows that CD4 count is reported here as the median and IQR for each year. The “N with data” column shows how many participants we have data for, total across the years. It is unclear if adding a separate table for number of participants for CD4 count and other variables per year would meaningfully add to the findings we report in the manuscript. Other reviewers did not find this confusing. Additionally, our data team has concerns with presenting a table without including the total N with data per variable.

It would be valuable to add a decision tree to this manuscript to graphically depict the number of CrAg positive (436), how many are on ART, have meningitis symptoms, etc. Is there any data as well on whether antifungal treatment was administered, which is a major gap in our data systems. Would it also be possible to add how many had a CD4 ≤200 cells/µL that were CrAg negative as well.

Response: Unfortunately, we do not have this level of information about the participants. Here we present the number of people screened CrAg positive, and titer by year. We only have data regarding antifungal treatment and how many have meningitis symptoms, for participants who were enrolled in clinical trials, which is a subset of the overall cohort. This is an important topic that would be best addressed in another submission as it out of the scope of this manuscript.

In Results, please indicate how many PLHIV were CrAg negative. In addition, the CrAg detection rate would be important for comparison purposes to other settings.

Response: This is an interesting question, but out of the scope of our manuscript. The primary question our manuscript seeks to address is the change in CrAg titer over time, not rates of CrAg positivity. Rates of CrAg positivity have been described in Uganda numerous times, and have remained stable at 5 to 8% among people with CD4<100 cells/µL. We have not collected data on the number of CrAg negative individuals (for reference, given the estimated 5% prevalence, this would require us to collect data on >8000 individuals, that are not the population of interest.

Figure 1 appears to report data by quarters for the period between 2019 and 2022. The legend refers to quartiles. Please report all four quarters for each year from 2017 to 2022. If this data is not available, perhaps report the data by year.

Response: We present data from November 2017 to May 2022. From 2017 to 2018 we only had 24 samples. Thus, presenting data by quarter for these 8 quarters results in very few data points. We thought it was reasonable to combine data from 2017-2018 (essentially 13 months of data).

Minor issues

In the abstract, please indicate that the CD4 threshold was a count ≤200 cells/µL. Please also indicate that advanced HIV disease in this study excludes clinical staging (I to IV).

Response: Specification of CD4 count added to abstract, see lines 33-34. This should make clear the parameters we are using in the study to define advanced HIV. We did not use clinical staging, but feel that it is odd to state that explicitly in the abstract. By presenting CD4 inclusion criteria, it is implied that clinical staging isn’t used. Clinical staging is not an accurate way to identify people with low CD4 count and would not be typically used where CD4 testing is available.

The abstract would read better if sub-sections were added for Introduction, Methods, Results and Conclusions.

Response: Thank you, sub-sections added to the abstract.

In the Background on line 54, please indicate the CD4 threshold for the CrAg screening program. Please also indicate whether this was a reflexed of provider-initiated approach.

Response: Thank you, this has been added to line 58-59.. We prefer not to comment on the method of screening, as we cannot confirm that every laboratory provided reflexive testing.

In the Methods in line 69 – CrAg is repeated in the same sentence.

Response: Thank you, this has been removed.

Line 85 – Please revise RStudio 2024 to ‘RStudio: Integrated Development Environment for R’

Response: Thank you, we have added this revision (lines 98-99).

Reviewer #3: The authors report an important analysis of cryptococcal antigen (CrAg) titers in Uganda during a period of expanded CrAg screening and ART scale-up. The observation that titers have not declined, and may even be increasing, despite these efforts is both notable and concerning. The authors convey a clear message that HIV remains a major public health challenge, the burden of advanced HIV disease (AHD) persists, and sustained efforts toward early detection and prevention remain essential.

I wonder whether increased screening efforts may have enabled the identification of a higher-risk population, particularly ART-experienced yet non-adherent patients (as possibly suggested in Table 1), which could partially explain the lack of overall improvement in findings. Despite the higher titers observed, it may be encouraging that these patients are being identified in the outpatient setting, reinforcing the importance of continued screening efforts.

The manuscript is well written, and I have no major concerns regarding the study design or analysis. The comments below are mostly to clarify several points and strengthen interpretation of the findings.

1. The authors state (Line 73): “In this analysis were asymptomatic adults with HIV who received a CrAg-positive result in plasma with an associated titer performed.” However, in Table 1, “symptoms of meningitis” are reported in a substantial proportion of the cohort (e.g., 42% overall).

It would be helpful to clarify this apparent discrepancy. Does “asymptomatic” refer specifically to the absence of meningitis symptoms at the time of screening, with symptoms assessed later during clinical evaluation?

Response: Thank you for this important clarification. We have removed the term asymptomatic. Typically, these are outpatients with advanced HIV disease found to be CrAg positive at screening. But as the reviewer notes, some have symptoms and other don’t.

2. The authors state (Lines 49 and 68): “The World Health Organization (WHO) recommends CrAg screening for adults with a CD4 cell count <100 cells/μL…” and “We prospectively screened adults with advanced HIV disease (CD4 <200 cells/μL)… to identify potential participants for clinical trials…”

It may be helpful to clarify which CD4 cutoff (100 vs 200) was applied for CrAg screening in the cohort included in this analysis. Additionally, please specify whether any further inclusion or exclusion criteria were used, given that participants were approached for clinical trials, and confirm whether these criteria remained consistent between 2017 and 2022.

Response: Thank you. Yes, initial recommendations were to screen those with CD4<100 cells/µL. Though this threshold has been raised (conditionally) to include all people with CD4<200 cells/µL. We have clarified this in the Background (lines 57-59), and the Methods (lines 79-82). There were no other inclusion/exclusion criteria used for this analysis. Here we present screening data of all those CrAg positive between 2017-2022. For clinical trials, there are other exclusion criteria, but those do not apply to this cohort.

Minor comments:

3. If available, it would be informative to report whether mortality changed over time. If mortality declined despite stable or increasing titers, this could suggest improvements in clinical management or the effectiveness of earlier identification and treatment following screening.

Response: Unfortunately, we don’t have mortality data on these individuals who were screened. We only have mortality data on participants who were enrolled, which is a subset of the overall cohort presented. Mortality among participants enrolled in clinical trials may not be representative or indicative of mortality among those screened and not enrolled in clinical trials.

4. Were there changes in routine CD4 testing practices (e.g., greater reliance on viral load monitoring) that may have delayed recognition of advanced immunosuppression and, consequently, eligibility for CrAg screening? This is more speculative but may help contextualize the findings.

Response: This is a very interesting point, and there is evidence that baseline CD4 testing rates diminished starting in 2013 with the World Health Organization’s implementation of “Treat all”. We have added the following to the discussion (lines 176-183)

“Of note, the adoption of the WHO “Treat All” strategy in Uganda, which recommends ART initiation for all people with HIV regardless of CD4 count, has led to diminished rates of CD4 testing. Baseline CD4 assessment for adults initiating ART decreased from 73% to 21% between 2013 and 2018. The “Treat All” strategy, while meant to reduce barriers to ART initiation and reduce morbidity and mortality among people with HIV, raises concerns that people with advanced HIV may not receive opportunistic infection screening and prophylaxis and could be a factor in the rise in CrAg titer over time even with increased access to ART.”

Reviewer #4: Review

Change in plasma cryptococcal antigen titers in Uganda among outpatients with advanced HIV disease from 2017 to 2022 with rollout of national cryptococcal screening

A very interesting concept for a manuscript, however the median quarterly CrAg titer from 2017-2022 demonstrated a non-statistically significant positive trend in CrAg titer (tau=0.385, p=0.086). Would any other factors affect enrolment in care and CrAg titres such as education, income/employment, marital status, sexual orientation etc? It seems like the COVID-19 pandemic and lockdown measures may have played a role in reducing patient enrolment/follow-up in care and access to antiretroviral therapy, probably resulting in patients presenting with more advanced disease? This should be discussed

Response: Unfortunately, we do not have this level of detail on screened individuals. Thank you for noting the impact of the COVID-19 pandemic, we have acknowledged the limitation of our study occurring during the period of the COVID-19 pandemic in the discussion (lines 209-213). We have added more context on access to antiretroviral therapy in the discussion (lines 154-158). The first paragraph of the Discussion now reads as follows:

“In this analysis of Ugandan adults with advanced HIV disease and cryptococcal antigenemia, median annual CrAg titer did not decrease between 2017 and 2022. In contrast, there was a modest positive trend in median annual titer, though not statistically significant. We had hypothesized that the CrAg titer would decrease given the availability of ART and rollout of the national CrAg screening program in Uganda. Since the implementation of PEPFAR in Uganda in 2004, access to ART has steadily increased. ART coverage in Uganda has ranged from 67% in 2017 to 80% in 2022. Despite this improvement, no change has been noted in CrAg titer over this time. Notably, in our cohort, 58% were ART naïve in both 2021 and 2022, the highest rates over our study period, suggesting ongoing late presentation to care. Thus, there is a disconnect between the national reports of improvement in ART coverage, and ART coverage of this population with advanced HIV disease.”

Methods

Study setting and patients

Page 4 Line 68-69 : We prospectively screened adults with advanced HIV disease (CD4<200) for CrAg in the plasma using the CrAg lateral flow assay…..

Were these patients newly enrolled in care? Defaulted and returned to care?

Response: Typically, individuals who get CD4 testing are either a) newly enrolled into care, or b) returning to care, or c) people who have defaulted their ART. We have added the following to lines 79-81):

“Those who received CD4 testing had either newly enrolled into care, were returning to care, or had ceased taking antiretroviral therapy (ART).”

Page 4 Line 72-73: Participants in this analysis were asymptomatic adults with HIV who received a CrAg-positive result…..

Here the adults were said to be asymptomatic but in Table 1, a number of participants had symptoms of meningitis. How is this explained?

Response: Thank you for this important clarification. This has been corrected and we no longer refer to them as asymptomatic.

Page 4 Line 76-78 : In May 2022, our clinical trial began only enrolling participants with high titers (>=1:160). Thus, we did not include participants screened after May 31, 2022 due to possible bias in screening practices ….

It would have been better not include patients screened in May 2022 as the clinical trial began in May 2022. A better cut-off may have been April 30, 2022

Response: Thank you. To clarify, the DSMB recommended that the clinical trial stopped screening participants with low titers on May 26, 2022. Therefore, we set May 31, 2022 as a cutoff date for inclusion in our analysis, given that there would likely be few participants screened in those 3 weekdays at the end of May. We have added the following in lines 89-92 to be more precise:

“On May 26 2022, our clinical trial stopped enrolling participants with low CrAg titers (<1:80). Thus, we did not include participants screened after May 31, 2022 due to possible bias in screening practices favoring high titer participants.”

Results

In Table 1, p values of 0.004 and 0.027 were obtained for the Plasma CrAg titer and Plasma CrAg Titer<1:160 respectively, but these have not been adequately explained

Was a multivariable analysis conducted?

Response: The results presented in Table 1 shows the trend of each variable over year. The P- values simply shows whether the particular variable measurement differs over years independently. Multivariable or co-factor adjusted analysis was not conducted.

The trend analysis of CrAg titers is summarized in the results section in lines 129-133. We used the Kendall’s tau coefficient to examine the trend in CrAg titer.

Discussion

Page 7 Line 140: First, health systems must have timely CD4 testing in place …

What method of CD4 testing is available in Uganda ? Is Visitect available

Response: CD4 availability is evolving over time. We have intentionally not commented on what is available today in Uganda, knowing that this may change in a month/year. Currently, a mixture of Visitect, and point-of-care CD4 technologies are available. Specifying what i

---

## [Decision Letter · Decision Letter 1]

26 Apr 2026

PONE-D-25-65883R1Change in plasma cryptococcal antigen titers in Uganda among outpatients with advanced HIV disease from 2017 to 2022 with rollout of national cryptococcal screeningPLOS One

Dear Dr. Rajasingham,

Thank you for submitting your manuscript to PLOS ONE. After careful consideration, we feel that it has merit but does not fully meet PLOS ONE’s publication criteria as it currently stands. Therefore, we invite you to submit a revised version of the manuscript that addresses the points raised during the review process.

We look forward to receiving your revised manuscript.

Kind regards,

Felix Bongomin, MB ChB, MSc, MMed, FECMM

Academic Editor

PLOS One

Journal Requirements:

Reviewers' comments:

Reviewer's Responses to Questions

**Comments to the Author**

1. If the authors have adequately addressed your comments raised in a previous round of review and you feel that this manuscript is now acceptable for publication, you may indicate that here to bypass the “Comments to the Author” section, enter your conflict of interest statement in the “Confidential to Editor” section, and submit your "Accept" recommendation.

Reviewer #2: All comments have been addressed

Reviewer #3: All comments have been addressed

Reviewer #4: (No Response)

2. Is the manuscript technically sound, and do the data support the conclusions?

Reviewer #2: Yes

Reviewer #3: Yes

Reviewer #4: Yes

3. Has the statistical analysis been performed appropriately and rigorously? 

Reviewer #2: Yes

Reviewer #3: Yes

Reviewer #4: Yes

4. Have the authors made all data underlying the findings in their manuscript fully available?

Reviewer #2: Yes

Reviewer #3: Yes

Reviewer #4: No

5. Is the manuscript presented in an intelligible fashion and written in standard English?

Reviewer #2: Yes

Reviewer #3: Yes

Reviewer #4: Yes

6. Review Comments to the Author

Reviewer #2: All aspects raised have been addressed in the revision of this manuscript. It would be interesting to received a subsequent analysis that includes additional outcomes such as the lumbar puncture data.

Reviewer #3: Overall, this is improved, clearer, and the authors addressed the earlier reviewer comments well. Below are a few minor comments to tighten methods clarity and interpretation.

1. Abstract, Methods (lines 30–31) “We prospectively screened adults with advanced HIV disease (CD4<200 cells/μl) for CrAg from 2017 through 2022 using the lateral flow assay and assessed median plasma CrAg titer.”

- It would be helpful to specify whether participants with prior cryptococcal meningitis were excluded, both in the Abstract and main Methods.

2. Abstract, Results (line 32) “From November 2017 to May 2022, 436 adults with advanced HIV disease had a positive plasma CrAg test.”

- Apologies if this was discussed already in the prior review, but is there a denominator and percent positive?

3. Lines 39–40: “Despite expanded access to ART nationally, 58% of study participants in 2022 were ART-naïve, suggesting ongoing late presentation to care for people with advanced HIV disease.”

- This is reasonable interpretation, but ART-experienced patients who still develop cryptococcosis may be an equally important signal of poor adherence, treatment interruption, virologic failure, or incomplete immune recovery. May consider broadening the interpretation to include both late presentation and ineffective long-term treatment or retention.

4. Methods (lines 80–82) “Participants in this analysis were adults with advanced HIV disease who received a CrAg-positive result in plasma with an associated titer performed.”

- Were individuals with a history of CM excluded?

5. The Table presents variables that are not described in the Methods. If accurate, state that age, sex, CD4, ART status, time on ART, symptoms, and titer were collected and analyzed. Would be helpful to specify which meningitis symptoms were assessed.

6. Discussion (lines 138–139) “We had hypothesized that the CrAg titer would decrease given the increased availability of ART and rollout of the national CrAg screening program in Uganda.”

- Another likely explanation may be that this is not just about ART access. It may also reflect adherence challenges, treatment interruption, stigma, retention failures, etc. If adherence data were not collected, the authors might consider acknowledging this as a limitation in interpreting the findings.

7. If screening volume or coverage increased over time, later years may reflect improved capture of high-risk patients rather than stable or worsening epidemiology alone. This may be hard to prove, but it may be a real potential bias that may be considered.

8. “Screening occurred at outpatient HIV clinics in the Kampala and Wakiso districts and the Masaka Medical Research Council field station…”

- Along the same lines, were these same catchment areas represented throughout the study period? If geographic coverage changed over time, that could alter case mix.

Reviewer #4: Change in plasma cryptococcal antigen titers in Uganda among outpatients with

advanced HIV disease from 2017 to 2022 with rollout of national cryptococcal

screening

This version is very much improved and I have only a few minor commets

Abstract

Page 2 Lines 38-39 : “Despite expanded access to ART nationally, 58% of study participants in 2022 were ART-naïve” ….These results are in the conclusion but are not in the results section of the abstract

Introduction

Page 3 Lines 61: “Elevated CrAg titer indicates higher risk of meningitis and death’… This seems to be a repeat of Lines 58-59

Pages 3-4 Lines 66-68: “We hypothesize that CrAg titer has decreased annually, as people are presenting to care earlier in their disease course before onset of fulminant meningitis”. However, the subjects enrolled in the study were either newly enrolled into care, returning to care, or had ceased taking ART. For the hypothesis to be valid, should the study patients be only those who are newly enrolled to care? Those participants returning to care, or had ceased taking ART are repeat participants to care

Results

A number of studies in Uganda indicate that men tend to present later for HIV care than women however in this study, the number of males and females were almost equal. Any reason for this?

Men’s late presentation for HIV care in Eastern Uganda: The role of masculinity norms | PLOS One

7. PLOS authors have the option to publish the peer review history of their article (what does this mean?). If published, this will include your full peer review and any attached files.

Reviewer #2: **Yes:** Naseem Cassim

Reviewer #3: No

Reviewer #4: No

---

## [Author Response · Author response to Decision Letter 2]

5 May 2026

Dear Dr. Bongomin,

Thank you for your further detailed review of our manuscript, “Change in plasma cryptococcal antigen titers in Uganda among outpatients with advanced HIV disease from 2017 to 2022 with rollout of national cryptococcal screening”. We have gone through the editor’s and reviewers’ comments in detail and have addressed all the editorial suggestions.

Below is our point-by-point reply to the reviewers’ comments in italics.

Reviewer #2: All aspects raised have been addressed in the revision of this manuscript. It would be interesting to receive a subsequent analysis that includes additional outcomes such as the lumbar puncture data.

Response: Thank you for your comments and suggestions, we will consider this for future projects.

Reviewer #3: Overall, this is improved, clearer, and the authors addressed the earlier reviewer comments well. Below are a few minor comments to tighten methods clarity and interpretation.

1. Abstract, Methods (lines 30–31) “We prospectively screened adults with advanced HIV disease (CD4<200 cells/μL) for CrAg from 2017 through 2022 using the lateral flow assay and assessed median plasma CrAg titer.”

- It would be helpful to specify whether participants with prior cryptococcal meningitis were excluded, both in the Abstract and main Methods.

Response: Thank you for your comment. We did not exclude people with a prior history of cryptococcal meningitis, as CrAg screening is routine. In this manuscript we summarize CrAg screening and titers by month and quarter. The inclusion criteria for the study is: adults with advanced HIV disease who received a positive CrAg plasma result with an associated titer performed (see lines 89-91). Given that we haven’t specified anything about prior cryptococcal meningitis, the reader can assume that we did not exclude participants with prior cryptococcal meningitis.

2. Abstract, Results (line 32) “From November 2017 to May 2022, 436 adults with advanced HIV disease had a positive plasma CrAg test.”

- Apologies if this was discussed already in the prior review, but is there a denominator and percent positive?

Response: No, we cannot present a denominator. We are not reporting prevalence of CrAg positivity here. We are reporting trends in titer over time. We do not have data on all CrAg negative individuals (who were thereby ineligible for our study).

3. Lines 39–40: “Despite expanded access to ART nationally, 58% of study participants in 2022 were ART-naïve, suggesting ongoing late presentation to care for people with advanced HIV disease.”

- This is reasonable interpretation, but ART-experienced patients who still develop cryptococcosis may be an equally important signal of poor adherence, treatment interruption, virologic failure, or incomplete immune recovery. May consider broadening the interpretation to include both late presentation and ineffective long-term treatment or retention.

Response: Thank you, this is an interesting point to consider, and we have added context to the manuscript regarding the implication of participants that are not ART-naïve presenting to care with a low CD4 count and positive CrAg result.

Please see lines 45-46 of the abstract:

“In addition, the presence of ART-experienced patients in our cohort suggests challenges with treatment adherence and retention.”

And, lines 169-186 and 269-271 in the discussion:

“Additionally, the presence of participants in our cohort that were ART-experienced and yet still presented with a CD4 count less than 200 and were CrAg-positive suggests challenges with treatment adherence, interruption of treatment, virologic failure, or incomplete immune recovery.”

“In addition, the presence of ART-experienced patients in our study population suggests ongoing challenges with treatment adherence and retention.”

4. Methods (lines 80–82) “Participants in this analysis were adults with advanced HIV disease who received a CrAg-positive result in plasma with an associated titer performed.”

- Were individuals with a history of CM excluded?

Response: No, individuals with a history of CM were not excluded if they met the inclusion criteria listed. We have not specifically mentioned this, because it was not an exclusion criterion.

5. The Table presents variables that are not described in the Methods. If accurate, state that age, sex, CD4, ART status, time on ART, symptoms, and titer were collected and analyzed. Would be helpful to specify which meningitis symptoms were assessed.

Response Thank you for your comment. We have specified the symptoms that were part of the assessment in the table caption:

“Symptoms of meningitis included one or more of the following: headache, nuchal rigidity, photophobia, confusion, focal neurologic deficits, mania, or fever.” – lines 139-141

Additionally, we have added the following sentence (lines 97-98) to the Methods:

“Baseline characteristics including age, sex, CD4 count, ART status, time on ART, and symptoms at presentation were collected and analyzed.”

6. Discussion (lines 138–139) “We had hypothesized that the CrAg titer would decrease given the increased availability of ART and rollout of the national CrAg screening program in Uganda.”

- Another likely explanation may be that this is not just about ART access. It may also reflect adherence challenges, treatment interruption, stigma, retention failures, etc. If adherence data were not collected, the authors might consider acknowledging this as a limitation in interpreting the findings.

Response: Yes, in our cohort, ART access was shown to be in decline, which we address in the discussion. See lines 164-168 of the discussion:

“Notably, in our cohort, 58% were ART naïve in both 2021 and 2022, the highest rates over our study period, suggesting ongoing late presentation to care. There was a statistically significant decline in prevalence of ART coverage at enrollment over time. Thus, there is a disconnect between the national reports of improvement in ART coverage, and ART coverage of this population with advanced HIV disease.”

Adherence data was collected, as we have data of participants that were currently on ART, never on ART, and previously on ART. See Table 1 on line 138.

We have added more context to the discussion regarding the implications of participants that are not ART-naïve presenting with a positive CrAg plasma result. Please see lines 169-186 and 269-271 in the discussion:

“Additionally, the presence of participants in our cohort that were ART-experienced and yet still presented with a CD4 count less than 200 and were CrAg-positive suggests challenges with treatment adherence, interruption of treatment, virologic failure, or incomplete immune recovery.”

“In addition, the presence of ART-experienced patients in our study population suggests ongoing challenges with treatment adherence and retention.”

7. If screening volume or coverage increased over time, later years may reflect improved capture of high-risk patients rather than stable or worsening epidemiology alone. This may be hard to prove, but it may be a real potential bias that may be considered.

Response: Thank you for this thoughtful comment. We have added this as a limitation in the Discussion (lines 225-227).

“If volume of screening changed over time, trends may reflect better documentation of titer or CrAg result, rather than worsening epidemiology”

8. “Screening occurred at outpatient HIV clinics in the Kampala and Wakiso districts and the Masaka Medical Research Council field station…”

- Along the same lines, were these same catchment areas represented throughout the study period? If geographic coverage changed over time, that could alter case mix.

Response: True. In our case, these same areas were represented throughout the study period.

Reviewer #4: Change in plasma cryptococcal antigen titers in Uganda among outpatients with

advanced HIV disease from 2017 to 2022 with rollout of national cryptococcal

screening

This version is very much improved and I have only a few minor commets

Abstract

Page 2 Lines 38-39 : “Despite expanded access to ART nationally, 58% of study participants in 2022 were ART-naïve” ….These results are in the conclusion but are not in the results section of the abstract

Response: Thank you, we have now added to the results section of the abstract, see lines 40-41.

“There was a statistically significant decline in the percentage of participants taking ART at the time of screening (p<0.001), with 58% reporting never having taken ART in 2022.”

Introduction

Page 3 Lines 61: “Elevated CrAg titer indicates higher risk of meningitis and death’… This seems to be a repeat of Lines 58-59

Response: Agreed, we have deleted this sentence in line 58-59.

Pages 3-4 Lines 66-68: “We hypothesize that CrAg titer has decreased annually, as people are presenting to care earlier in their disease course before onset of fulminant meningitis”. However, the subjects enrolled in the study were either newly enrolled into care, returning to care, or had ceased taking ART. For the hypothesis to be valid, should the study patients be only those who are newly enrolled to care? Those participants returning to care, or had ceased taking ART are repeat participants to care

Response: Thank you for this thoughtful comment. Our study population includes those with advanced HIV. The proportion newly engaged in care vs. returning to care is unknown, and quite difficult to discern. Our hypothesis was that the expansion of ART availability and the national CrAg screening program would lead to presentation to care earlier in the disease course. This might be true for ART naïve individuals, but also for ART experienced individuals. Perhaps with expanded access to ART, people who are ART-experienced are less likely to stay out of care for so long. Or are more likely to resume care and get CrAg screened.

Results

A number of studies in Uganda indicate that men tend to present later for HIV care than women however in this study, the number of males and females were almost equal. Any reason for this?

Men’s late presentation for HIV care in Eastern Uganda: The role of masculinity norms | PLOS One

Response: This is a good question. While men may present to HIV care later and with more advanced disease, they will eventually reach care in similar numbers. This may be a reason why males and females are represented in equal numbers in our study. It is possible that men presented with more advanced HIV disease. While interesting, that is not the purpose of this study, so we have not pursued this further.

All authors have approved of the manuscript. The manuscript is not under consideration elsewhere and has not been previously published. There are no conflicts of interest by any author.

Warmly,

Radha Rajasingham, MD

Associate Professor

Infectious Diseases & International Medicine

Department of Medicine, University of Minnesota

---

## [Editor Report · Decision Letter 2]

6 May 2026

Change in plasma cryptococcal antigen titers in Uganda among outpatients with advanced HIV disease from 2017 to 2022 with rollout of national cryptococcal screening

PONE-D-25-65883R2

Dear Dr. Rajasingham,

We’re pleased to inform you that your manuscript has been judged scientifically suitable for publication and will be formally accepted for publication once it meets all outstanding technical requirements.

Kind regards,

Felix Bongomin, MD

Academic Editor

PLOS One
---

## [Editor Report · Acceptance letter]

PONE-D-25-65883R2

PLOS One

Dear Dr. Rajasingham,

I'm pleased to inform you that your manuscript has been deemed suitable for publication in PLOS One. Congratulations! Your manuscript is now being handed over to our production team.

Kind regards,

on behalf of

Dr. Felix Bongomin

Academic Editor

PLOS One